# Settlement and Stress Characteristics of the Ground in the Project of a Double-Line Tunnel Undercrossing an Airport Runway in a Sandy Cobble Region

**Xuwei Zhao [1], Jia Li [2,3,*], Wei Liu [2,3] and Wenge Qiu [1]**

[1] Key Laboratory of Transportation Tunnel Engineering, Southwest Jiaotong University, Chengdu 610000, China
[2] Key Laboratory of Road and Traffic Engineering of the Ministry of Education, Tongji University, Shanghai 200032, China
[3] Shanghai Key Laboratory of Rail Infrastructure Durability and System Safety, Shanghai 200032, China
[*] Correspondence: 2111512@tongji.edu.cn

**Abstract:** The engineering technology of undercrossing an airport flight area is relatively mature; however, the use of shield tunnels crossing the operating airport flight area in high liquidity sandy cobble stratum has rarely been reported. To discuss the feasibility of a double-line shield tunnel undercrossing the airport flight area in a sandy cobble region. Based on the case of the Chengdu Metro Line 10 that undercrosses the Shuang-Liu Airport, which is located in sandy cobble region, the deformation and stress laws of airport runway pavement structures were investigated via a three-dimensional numerical model. Stratum displacement, ground settlement and pavement tensile stress under different tunnel depths were analysed. Then, the pavement tensile stress was taken as the safety evaluation index and the vulnerable area of the pavement structure was proposed. The results show that after excavation of the double-line tunnel, the maximum ground surface settlement occurred above the tunnel centreline and the ground settlement curves presented a V-type settlement trough instead of a W-type. The existence of a runway greatly limited the deformation of the surrounding soil; with increasing depth, the effect degree of the runway pavement on the soil settlement decreased. The most unfavourable region of the runway pavement structure under the influence of tunnel excavation was found. With increasing burial depth, the maximum settlement of the surface centre point decreased continuously. It is recommended that the tunnel burial depth not exceed 23.5 m in this project. According to the displacement and stress control limitation of the airport pavement, it can be judged that the shield construction method meets the structural stability requirements of the pavement. The calculated results provide a reference for the shield tunnel construction in the same geological condition areas.

**Keywords:** shield tunnel; undercrossing construction; settlement; stress; runway pavement

## 1. Introduction

With the rapid development of China's economy, the demand for travel is increasing. Zero distance transfer and seamless connections have become important factors in people's choice of travel mode. The rapid transportation network that is composed of railway, rail transit and aircraft has entered the rapid construction period. The complex three-dimensional transportation network has gradually formed, and some more complex transportation construction projects have emerged, such as the construction of underpass tunnels or other underground buildings in the airport flight area (runway, airport taxiway, etc.). At present, the existing domestic and foreign tunnels under the airport airfield include: Taiwan Songshan municipal road underground runway crossing project, Taipei shield tunnel crossing runway engineering, Beijing Capital Airport underground service lane undercrossing l-shaped slideway project and the runway crossing project of Shanghai Hongqiao airport metro Lines 2 and 10, plus, the circular monorail tunnel under the

Stuttgart Airport runway in Germany and the rail tunnel under the taxiway and apron of Heathrow Airport in the UK.

In contrast to other shield undercrossing engineering projects, the normal operation of the airport runway cannot be interrupted during the construction of the shield tunnel. Additionally, airport runway construction has more stringent requirements on ground settlement. Excessive ground displacement induced by underground crossings can cause runway cracking; this phenomenon may cause airport outages or major accidents [1,2]. So, excavating shield tunnels in non-stop airport flight areas is extremely challenging. It is necessary to carry out safety analysis and formulate a strict control scheme before construction. At present, much research has been performed on the settlement of the existing buildings and structures caused by shield tunnel construction. However, there is no clear and unified regulation on the allowable settlement of airport runways. The settlement control requirements of airport runways are mainly based on the experience of highway construction on soft soil foundations. There are few studies on the laws of airport runway settlement caused by an undercrossing shield tunnel.

Aiming at the issue of ground surface settlement caused by shield tunnel construction, the research methods can be summarized as follows: empirical formula method, numerical analysis method, theoretical analysis method and model test method. In terms of the empirical formula method, Peck [3] proposed a formula fitted by a Gaussian function regarding ground settlement induced by a tunnel. The Peck formula is widely accepted in tunnel engineering. On the basis of Peck's empirical formula, researchers have conducted more in-depth research on empirical parameters [4–8]. Compared with the empirical formula method, the theoretical analysis method is a conclusion analysis method based on some theoretical assumptions. Taking into account that the soil is elastic plane and incompressible, Sagaseta [9] proposed the strain solution of an elastic isotropic homogeneous body in the case of near-surface ground loss. Subsequent scholars solved the formation deformation under different boundary conditions through a complex potential function method [10–13], stochastic medium theory [14–17] and cavity expansion theory [18–21], etc.

With the rapid development of computer techniques, finite element numerical simulation is becoming the most efficient way to simulate the construction procedure of underground engineering under various complex environmental conditions [22–24]. For instance, Shan et al. [25] investigated the effects of shield tunnel construction on the differential settlement and the soil dynamic stress of an adjacent culvert-embankment transition zone via finite element analysis. Fattah et al. [26–29] investigated the shape of the settlement trough caused by tunnelling in cohesive ground by analytical solutions, empirical solutions and the finite element method. Chakeri et al. [30] established a three-dimensional numerical model to analyse the effect of tunnel spacing, horizontal distance between tunnels and excavation sequence on surface settlement. Michael et al. [31] simulated the shield tunnel mechanical excavation process including variable muck pressure, cutterhead overcut, shield conicity, installation of jointed segmental lining, annular gap grouting based on ABAQUS. Then, the development of ground deformations and internal forces of the lining were studied.

In the relevant study of undercrossing the airport runway, Tong et al. [32] analysed the stress and deformation of runway pavement induced by undercrossing tunnel excavation based on the mechanics method, and the calculation method of the soil settlement trough limit value and the vault control value were proposed. Shen and Elbaz [33] presented a novel deep learning model for real-time prediction of shield moving trajectory during tunnelling, and a tunnel case study at Bao'an International Airport, Shenzhen, was employed to demonstrate the accuracy of the proposed model. The field observation is the most popular and straightforward method for understanding the interaction between existing buildings and undercrossing tunnels. Huang et al. [34] introduced a super-large diameter earth pressure balance (EPB) shield tunnel crossing below an operating airport in a soft soil region and the environmental effect of EPB shield tunnelling was investigated by monitoring the surface settlement. Zhou et al. [35] studied the control rules of tunnel shield

construction parameters through field tests; the corresponding control measurements were given to reduce the settlement of airport taxiways, which provided guidance for construction parameter settings during shield advancing. Based on the observation data of runway surface settlements for more than 6 years, Yang et al. [36] studied the settlement law of airport runways under aircraft take-off and landing loads.

Different from the relatively mature technical conditions of shield tunnels undercrossing airports in coastal soft and clay areas, Chengdu Shuang-Liu Airport is located in a sandy cobble stratum with rich water and high mobility. The characteristics of this stratum are unstable mechanics, large gaps between particles, no cementation, sensitive response and ease of producing disturbance loss. When the cutter head is rotating and cutting, the stratum that is relatively stable or in equilibrium and is easily destroyed and then collapses, resulting in large stratum loss and surrounding rock disturbance, which has brought great challenges to the construction of metro shield tunnels that undercross airports. In addition, there are few reports of shield tunnels undercrossing operating airports in a high-flow sandy cobble stratum.

Therefore, it is necessary to conduct in-depth research on ground settlement and discuss the feasibility of shield tunnel undercrossing airport in sandy cobble area. Considering above problems, based on the Chengdu Metro Line 10 project undercrossing Shuang-Liu Airport, the influence law of double-line shield tunnel construction on the ground settlement was investigated via a numerical simulation method. Finally, the vulnerable area of the pavement structure was proposed through stress analysis and the shallowest tunnel buried depth in this region was determined. It was expected to make a rational calculation and assessments of the safety of the airport runway and provide reference for similar projects in the future.

## 2. Background

This study is based on the Chengdu subway project in China. The tunnel from Shuang-Liu West Station to Shuang-Liu Airport Terminal 2 of Chengdu Metro Line 10 is located in the Shuang-Liu District (Figure 1), which highlights the Chengdu Metro Line 10 Phase II Project undercrossing the Chengdu Shuang-Liu Airport flight area. The undercrossing area is mainly runways and sward, and the subway terminal station is Shuang-Liu Airport Terminal 2. The lithological log around the shield tunnels mostly consists of miscellaneous fill, silty clay, sandy cobble and weathered mudstone with increasing depth (Figure 2).

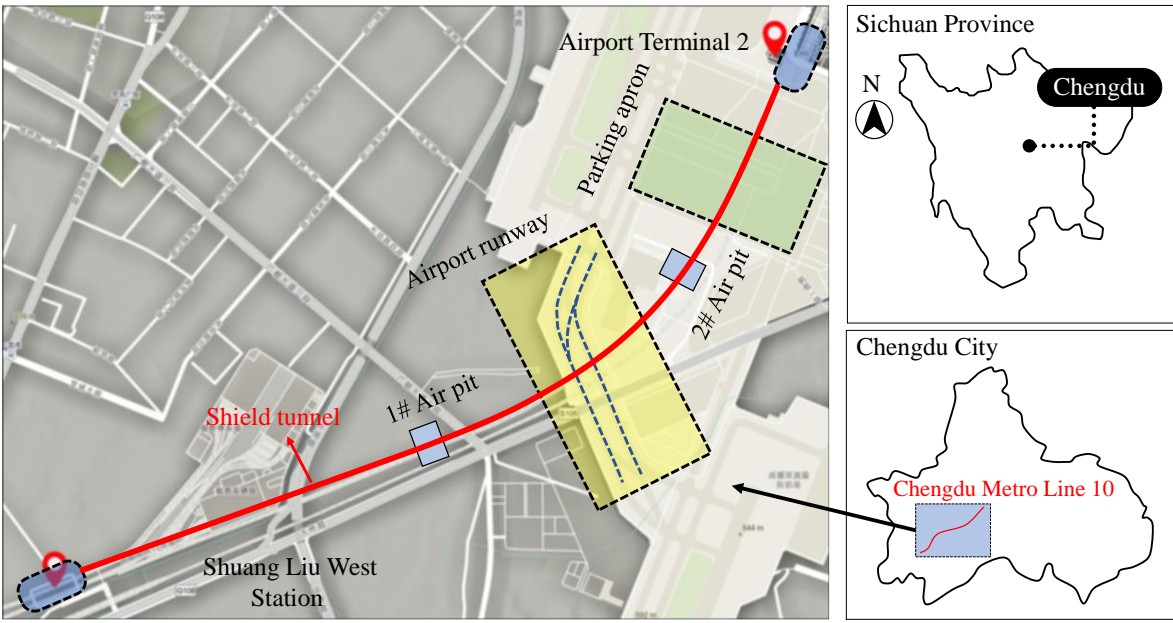

**Figure 1.** The project location map.

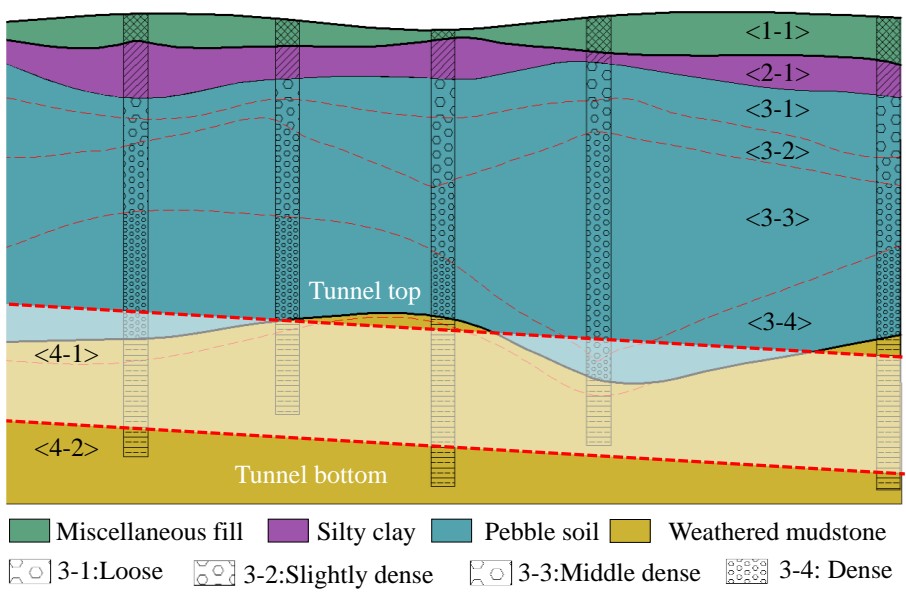

**Figure 2.** Geological profile.

The plan view and spatial location of the project are shown in Figure 3. The distance between the left tunnel and right tunnel is 13.0 m. The airport runway is a double line with a width of 60 m, and the width of the sward between the runways is 40 m. The undercrossing tunnel constructed by the shield-driven method is a conventional circular section with single-layer reinforced concrete precast lining. The burial depth of the tunnels is approximately 46.5 m. The thickness of the segment is 300 mm, the inner diameter is 5.4 m and the outer diameter is 6.0 m.

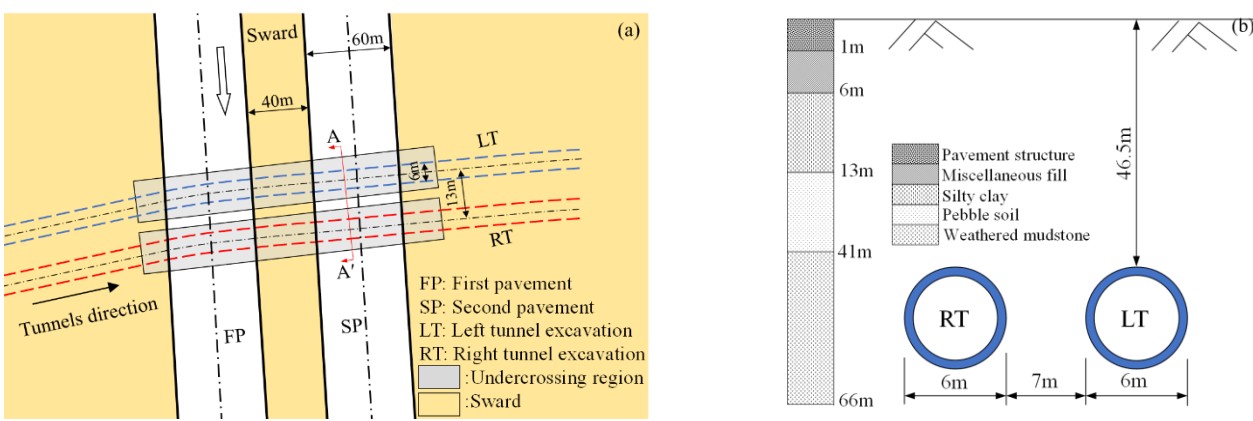

**Figure 3.** Illustration of excavated tunnels undercrossing airport runway: (**a**) plan view; (**b**) typical cross-section A-A′.

The typical soil profile at the site is presented in Figure 3b. The geological soil layer of the undercrossing of the airport runway section from top to bottom is pavement structure, miscellaneous fill, silty clay, pebble soil and strongly weathered mudstone. The shield tunnel is located in the weathered mudstone. The main physical and mechanical parameters of each layer are shown in Table 1.

The pebble soil and weathered mudstone strata are widely distributed in the Chengdu area; the special engineering characteristics of these strata have caused great difficulties for shield construction, as shown in Figure 4. The characteristics of these two strata are introduced in this section. The pebble stratum is brown grey or grey yellow, mainly loose or medium dense. The pebbles are mainly slightly weathered magmatic rocks and metamorphic rocks. The shape of pebbles is subcircular. The pebble content in this stratum

is generally 60~70%, the particle size is mainly 2~15 cm and the maximum particle size is approximately 40 cm. The filling materials are mainly fine and medium sand and round gravel. A pebble stratum is a typical mechanically unstable formation with a large gap between particles and almost no cohesion. Under anhydrous conditions, force is transmitted through point contact between particles and the stratum reaction is sensitive. During the construction of shield tunnels, the stable state of the stratum is easily destroyed and collapses, resulting in large stratum loss and disturbance to the surrounding rock.

**Table 1.** Physical and mechanical parameters of the soil.

| Soil Layer Number | Unit Weight $\gamma$ (kN/m$^3$) | Cohesion c (KPa) | Internal Friction Angle $\phi$ (°) | $E_{s0.1-0.2}$ (MPa) | Poisson's Ratio $v$ |
|---|---|---|---|---|---|
| Miscellaneous fill | 19.0 | 10.0 | 8 | / | 0.32 |
| Silty clay | 20.0 | 40.9 | 19 | 5.7 | 0.30 |
| Pebble soil | 22.0 | 20.0 | 40 | 20 | / |
| Weathered mudstone | 23.5 | 300.0 | 35 | 23 | 0.18 |
| Test | Routine physical test | Direct shear test | | Compression test | |

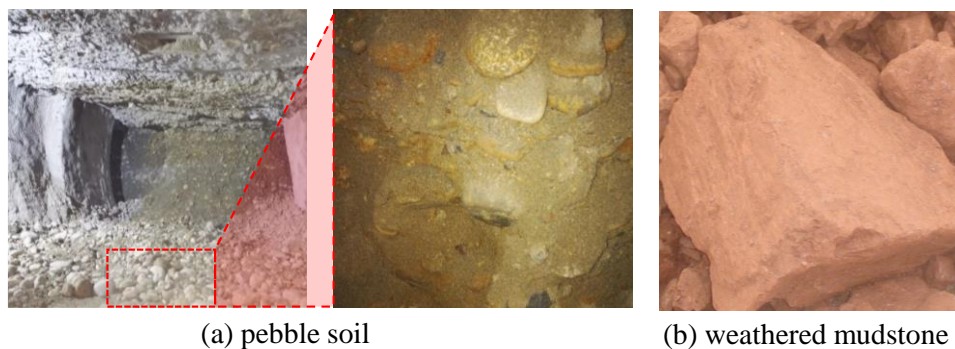

(a) pebble soil  (b) weathered mudstone

**Figure 4.** Soil layer of the tunnel excavation face.

The weathered mudstone stratum is dark red or purple red. The development degree of the rock joint fissure is relatively great and the rock quality designation (RQD) is 75~90%. The natural single axis compressive strength is in the range 3.15~6.70 MPa, the saturated single axis compressive strength is in the range 1.76~3.91 MPa, the free expansion rate is in the range of 12~28% and the expansion force is in the range 5.8~25.1 kPa.

In view of the particularity of sandy pebble strata, a project with a shield tunnel crossing an operating airport in a sandy pebble area has not been reported. Based on the actual project in Chengdu and the numerical simulation test, this paper discusses the feasibility of shield tunnels undercrossing airports in sandy pebble areas, which can provide some technical support for shield construction in other sandy pebble areas. The research scheme is shown in Figure 5.

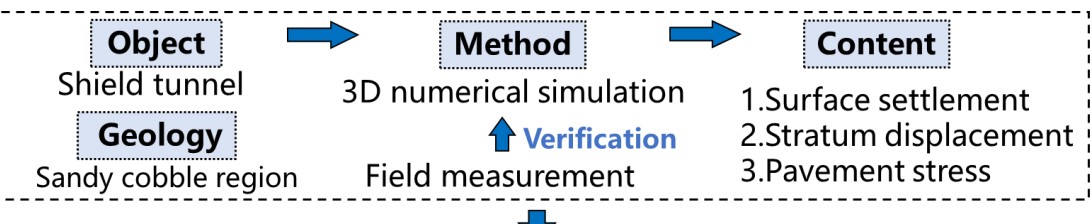

**Figure 5.** Research flow chart.

## 3. Three-Dimensional Numerical Model

### 3.1. FEM Model

A three-dimensional numerical model was established using the finite element software *ANSYS* to simulate the construction procedure of the shield tunnel. Figure 6 shows the finite element model used in this study, which was 100 m in the $x$ direction, 70 m in the $y$ direction and 200 m in the $z$ direction. The tunnel was 200 m long, 5.4 m in inner diameter, 6.0 m in outer diameter and 13 m in tunnel spacing. The widths of the runway pavement and sward were 60 m and 40 m, respectively, and the thickness of the pavement was 1 m. As presented in Figure 6, the Solid 45 element was used to simulate the soil and the shell element Shell 63 was used to simulate the shield shell and lining. The allowable displacement boundary conditions were as follows: no horizontal displacement along the full vertical mesh boundaries, no vertical and horizontal displacement along the bottom boundary of the mesh and free displacement along the top boundary.

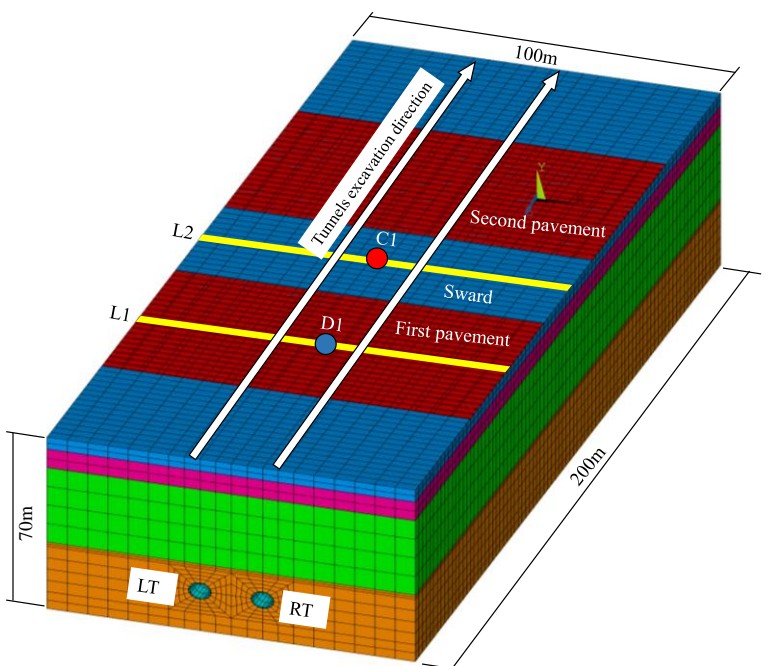

**Figure 6.** Three-dimensional numerical model of the shield tunnel.

### 3.2. Material Model and Parameters

The stratum, grouting material, shield shell and lining were simulated using three-dimensional solid elements. For the grouting material, shield shell and lining, the elastic model was adopted. For soil, the Mohr-Coulomb model was adopted, and the material parameters used in this study are shown in Tables 1 and 2. The binding constraints were set at the interface of the soil-shield shell, soil-grout and grout-lining. As shown in Figure 7, the runway pavement structure consisted of a concrete layer and a pebble layer. In this model, the pavement structure was simplified as linear elastic materials with an elastic modulus of 10,000 MPa. The groundwater infiltration and soil consolidation creep were not considered in this model.

**Table 2.** Material parameters in the model.

| Materials | Density (g/cm³) | Elastic Module (MPa) | Poisson Ratio |
|---|---|---|---|
| Pavement | 2.5 | 10,000 | 0.25 |
| Shield shell | 2.5 | 30,000 | 0.25 |
| Initial grouting | 2.1 | 3 | 0.30 |
| Hardening grouting | 2.1 | 20 | 0.30 |
| Lining | 14.2 | 20,000 | 0.30 |

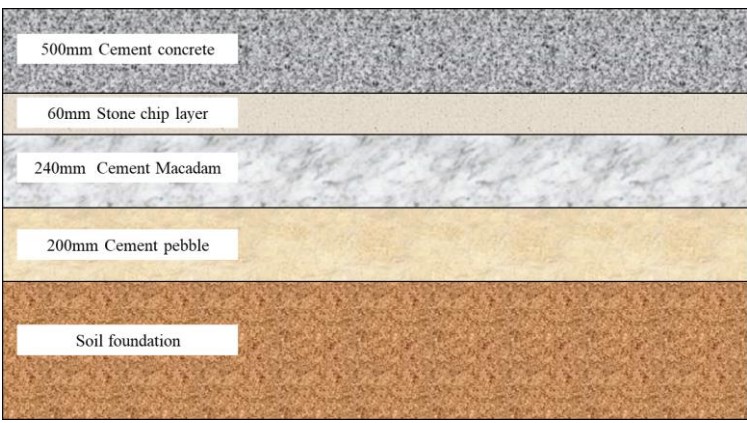

**Figure 7.** Runway pavement structure.

### 3.3. Simulation of the Excavation Process

In the simulation of the tunnel excavation, an excavation step was set as twice the duct piece width (2.4 m). The simulation of soil excavation, shield tunnelling, segment installation and synchronous grouting were realized via the element birth–death method; the excavation simulation diagram is shown in Figure 8. The detailed process is as follows:

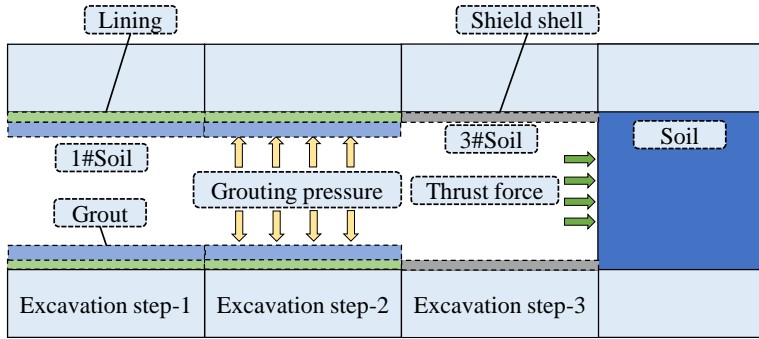

**Figure 8.** Simulation process of shield excavation.

STEP 1: Thrust was applied on the #1 soil element, and then the #1 soil element was removed. At the same time, the corresponding shield shell unit was activated.

STEP 2: The #2 soil element was removed. At the same time, the corresponding shield shell unit was activated. The shield shell unit in excavation step 1 was removed, the corresponding lining and grouting unit were activated and circumferential grouting pressure was applied to the surrounding soil. In the subsequent excavation steps, the above two steps were repeated until a total of 84 excavation steps were completed.

## 4. Settlement Caused by Shield Tunnel Excavation

To explore the influence of the runway pavement structure on ground surface settlement, three conditions were set for simulation. (1) Condition 1: shield tunnel undercrossing a double runway. (2) Condition 2: shield tunnel undercrossing a single runway. (3) Condition 3: shield tunnel undercrossing sward without runway. The accuracy of the model was verified by comparing the simulated and measured ground surface settlement under condition 1 and the simulations for conditions 2 and 3 were conducted based on this model. Thirteen surface settlement monitoring points were arranged by the hole-drilling method, and the distribution layout is shown in Figure 9. First, a Φ120 mm hole was opened in the ground and a Φ20 mm ribbed rebar was inserted. The length of the ribbed rebar was 1 m. Then, fine sand was filled around the rebar. The top of the fine sand was 5 cm away from the ground surface, the rebar was exposed to the fine sand for 1 cm and then a protective cover was used to cover the exposed steel bar.

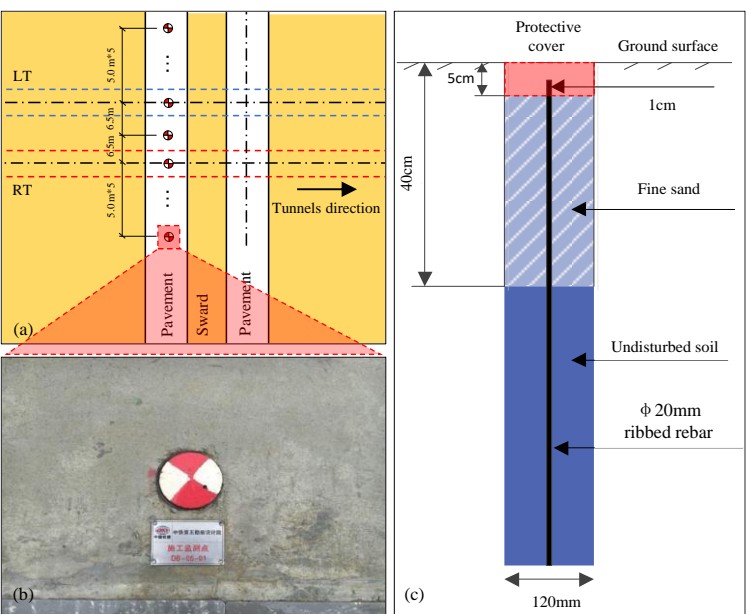

**Figure 9.** Distribution layout of monitoring points: (**a**) distribution of monitoring points; (**b**) site map of settlement monitoring points; (**c**) schematic diagram of monitoring points by hole-drilling method.

The comparison result of the simulated ground surface settlement and measured ground surface settlement for condition 1 is shown in Figure 10. It can be concluded that the measured settlement was less than the simulated settlement because of the influence of the complex construction environment on settlement was not considered in the model. However, the simulated settlement trend was similar to that of the measured settlement, which verified the accuracy of the model.

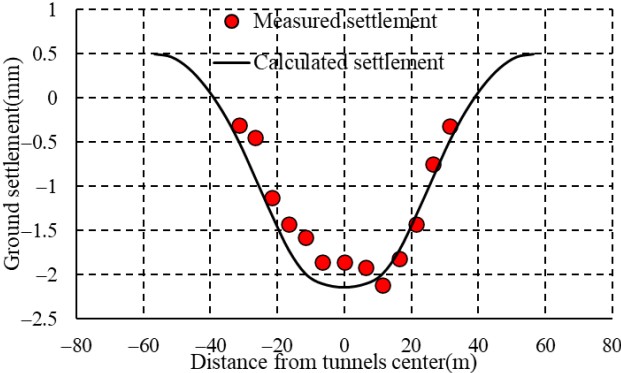

**Figure 10.** Calculated settlement vs. measured settlement.

### 4.1. Ground Surface Settlement Trough

Based on the proposed model, the influence of the double-line tunnel excavations on ground surface settlement was investigated. When the double-tunnel excavation was finished, the ground surface settlement curves along the runway pavement longitudinal section are shown in Figure 11. L1 and L2 represent the sections of the runway midline and sward midline, RL1′ and RL2′ are the sections of runway midline and sward midline when the right-line tunnel excavation was finished and LL1′ and LL2′ are the sections of runway midline and sward midline when the left-line tunnel excavation was finished, respectively. The ground surface settlement curves of RL1′, LL1′, RL2′ and LL2′ were normally distributed and the Peck formula was used to fit the settlement trough. As shown in Figure 8, there was a good fit between the settlement trough and the Peck formula. The settlement of the monitoring points above the tunnel axis was maximum. The farther the

monitoring point from the shield tunnel axis, the smaller the ground surface settlement. The ground surface settlement induced by a double-line shield tunnel excavation was always larger than that induced by a single-line shield tunnel excavation for different sections.

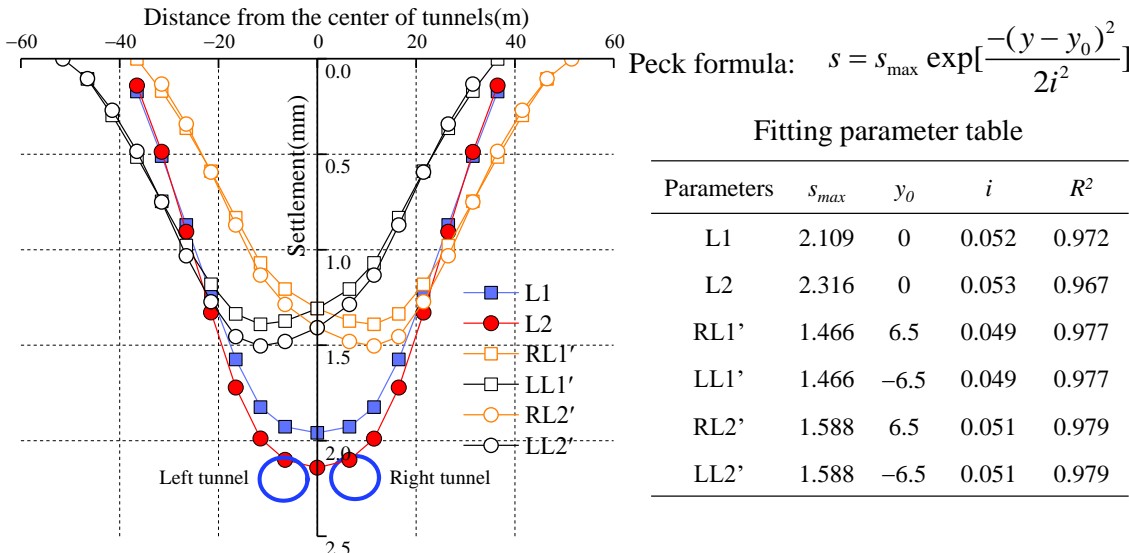

Peck formula:

$$s = s_{\max} \exp[\frac{-(y - y_0)^2}{2i^2}]$$

Fitting parameter table

| Parameters | $s_{max}$ | $y_0$ | $i$ | $R^2$ |
|---|---|---|---|---|
| L1 | 2.109 | 0 | 0.052 | 0.972 |
| L2 | 2.316 | 0 | 0.053 | 0.967 |
| RL1' | 1.466 | 6.5 | 0.049 | 0.977 |
| LL1' | 1.466 | −6.5 | 0.049 | 0.977 |
| RL2' | 1.588 | 6.5 | 0.051 | 0.979 |
| LL2' | 1.588 | −6.5 | 0.051 | 0.979 |

**Figure 11.** Ground surface settlement trough at different sections.

The ground surface settlement curves at different sections induced by a double-line tunnel undercrossing were V-type settlement troughs instead of W-type. A previous study showed that the type of settlement trough induced by tunnel undercrossing is closely related to the tunnel spacing (L), depth (H) and radius (R). When L/(H + R) ≤ 0.5, the settlement curves are V-type; in this study, L/(H + R) = 0.255, which was consistent with the V-type settlement trough characteristic. Among them, the maximum settlement at the L1 section was 1.96 mm, and the maximum settlement at the L2 section was 2.14 mm. This is because the runway pavement structure has a greater stiffness compared with the soil, so the runway itself had a small settlement.

### 4.2. Settlement of the Characteristic Point

The settlement variation of the characteristic monitoring point (D1) on the runway pavement and characteristic monitoring point (C1) on the sward with excavation steps is shown in Figure 12. When the shield machine advanced to the 40th loop, the settlement of D1 reached the maximum value of 1.96 mm. When the shield machine advanced to the 60th loop, the settlement of C1 reached the maximum value of 2.14 mm. After excavation step 40, the settlement increased slightly and the rising value basically remained at $1 \times 10^{-3}$ mm. Similarly, after excavation step 60, there was a very small decrease of the settlement and the rising value basically remained at $0.5 \times 10^{-3}$ mm. The magnitude of the increase was very small. Basically, it can be considered that the settlement is stable and the small amplitude increase of this settlement is probably caused by the formation of the shield shell and lining segment.

By comparing the simulated settlement with the measured settlement, the accuracy of the finite element model was verified. This can provide some guidance for settlement prediction in the construction process.

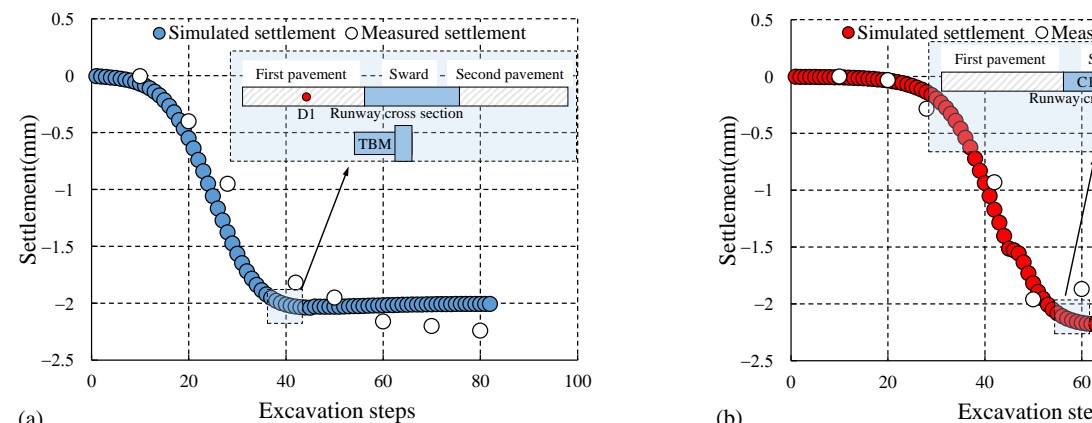

**Figure 12.** Settlements of characteristic points (**a**) D1 and (**b**) C1.

### 4.3. Stratum Displacement at Different Depths

To investigate the stratum displacement at different stratum depths affected by tunnel excavation, the settlement curves of the sward midline cross section at the ground surface and 15 m, 30 m and 45 m below the ground surface were calculated by the established model, as shown in Figure 13. It can be concluded that the ground settlement above the tunnel increased with the stratum depth and the maximum stratum settlement occurred above the tunnel axis. For condition 1, the maximum settlement of the ground surface was 2.14 mm and the maximum settlement of the stratum 45 m below the ground surface was 4.30 mm. For these conditions, the settlement curves of the cross section in the shallow stratum were V-type troughs. However, the settlement curves of the stratum near the tunnel top were W-shaped troughs and the maximum settlement occurred above the left and right tunnels.

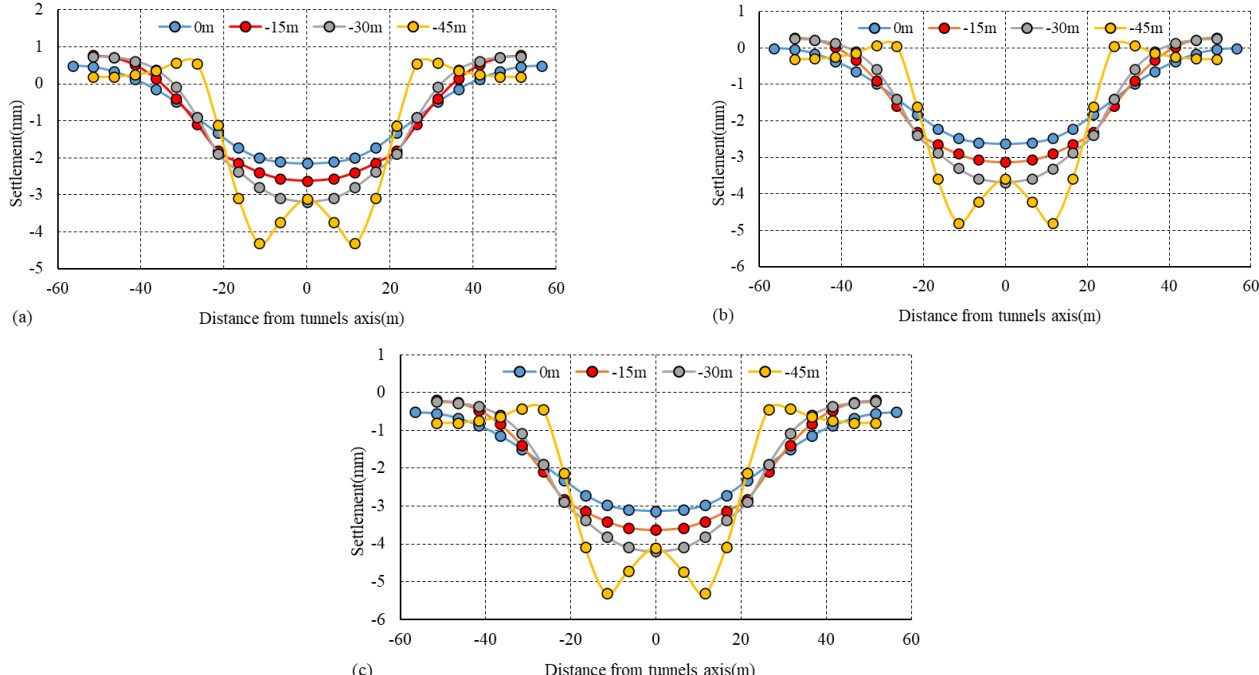

**Figure 13.** Settlement of soil layers at different depths under different conditions (**a**) condition 1, (**b**) condition 2 and (**c**) condition 3.

With increasing stratum depth, the trends of the settlement curves under the three conditions were similar; that is, the existence of a runway pavement structure had little

influence on the stratum settlement law. Comparing the maximum settlement of stratum at different depths under these conditions, it can be found that the maximum ground surface settlements for conditions 1, 2 and 3 were 2.14 mm, 2.63 mm and 3.13 mm, respectively, and the existence of a runway pavement structure can obviously decrease the ground surface settlement. This is because the settlement of the runway structure with larger rigidity compared with the soil is small and the existence of the runway can greatly limit the deformation of the surrounding soil, thus reducing the ground settlement.

Comparing the maximum settlement difference of the soil layers at different depths under conditions 1, 2 and 3, the settlement differences between the soil layers at different depths were 0.98, 0.81, 0.60 and 0.39 mm. The maximum settlement difference between conditions gradually decreased with increasing stratum depth, indicating that the influence of the runway layer on the ground settlement gradually decreased with increasing stratum depth. The existence of the runway structure was mainly manifested by the obvious increase in surface stiffness, the settlement of the surface and upper soil was reduced and the settlement difference between different conditions was mainly in the surface and upper soil.

### 4.4. Ground Surface Settlement for Different Burial Depths

Based on the proposed numerical model, to guide tunnel construction the minimum tunnel burial depth when the surface settlement does not exceed the limit was proposed and the influence of the tunnel burial depth H = 15.0 m, 22.5 m, 30.0 m, 43 m and 46.5 m on the ground surface settlement was investigated. The ground surface settlement curves along the runway pavement longitudinal section for different tunnel burial depths are shown in Figure 14a. The ground surface settlement trough under different tunnel burial depths was approximately consistent with the Peck formula and the tunnel burial depth had a significant effect on the surface settlement. When the burial depth was 22.5 m, the maximum surface settlement was 10.35 mm, and when the burial depth was 15.0 m, the maximum surface settlement was 15.09 mm. According the limit value of pavement settlement proposed by the previous literature [35] (settlement difference between construction area and surrounding area is no more than 10 mm), it is suggested that the buried depth of the shield tunnel in this project should be no greater than 22.5 m. Figure 14b shows the variation in the maximum surface settlement with different tunnel burial depths, which basically conformed to the exponential curve law. This shows that with increasing tunnel burial depth, the maximum settlement of the surface centre point decreased continuously but the decreasing trend gradually slowed down. Given the variation of runway pavement characteristic monitoring point (D1) settlement with excavation steps for different tunnel buried depths, it can be concluded that for a buried depth of 46.5 m, when the shield machine advanced to the 40th loop, the settlement of D1 reached the maximum value of 1.96 mm. For a buried depth of 15.0 m, when the shield machine advanced to the 70th loop, the settlement of D1 reached the maximum value of 15.09 mm. This shows that with the increasing shield tunnel buried depth, the development time of the ground surface settlement increased as well.

### 4.5. Pavement Structure Stress for Different Tunnel Burial Depths

During the tunnel excavation period, the ground surface settlement and pavement structure stress will change. In this paper, the finite element numerical model was used to investigate the pavement structure stress characteristics caused by the tunnel excavation under different tunnel burial depths. According to the distribution of pavement tensile stress, the pavement vulnerable areas were proposed for reference of pavement maintenance.

Case 1: the tunnel buried depth = 46.5 m.

The stress in the $x$ direction of the runway in the transverse section when the buried depth = 46.5 m is shown in Figure 15. The pavement within 29.8 m of the shield tunnel axis was subjected to compressive stress. When the pavement was 56.5 m away from the tunnel

axis, the stress basically dissipated to 0 kPa and the pavement 29.8–56.5 m away from the shield tunnel axis was subjected to tensile stress; the maximum tensile stress was 6.82 kPa.

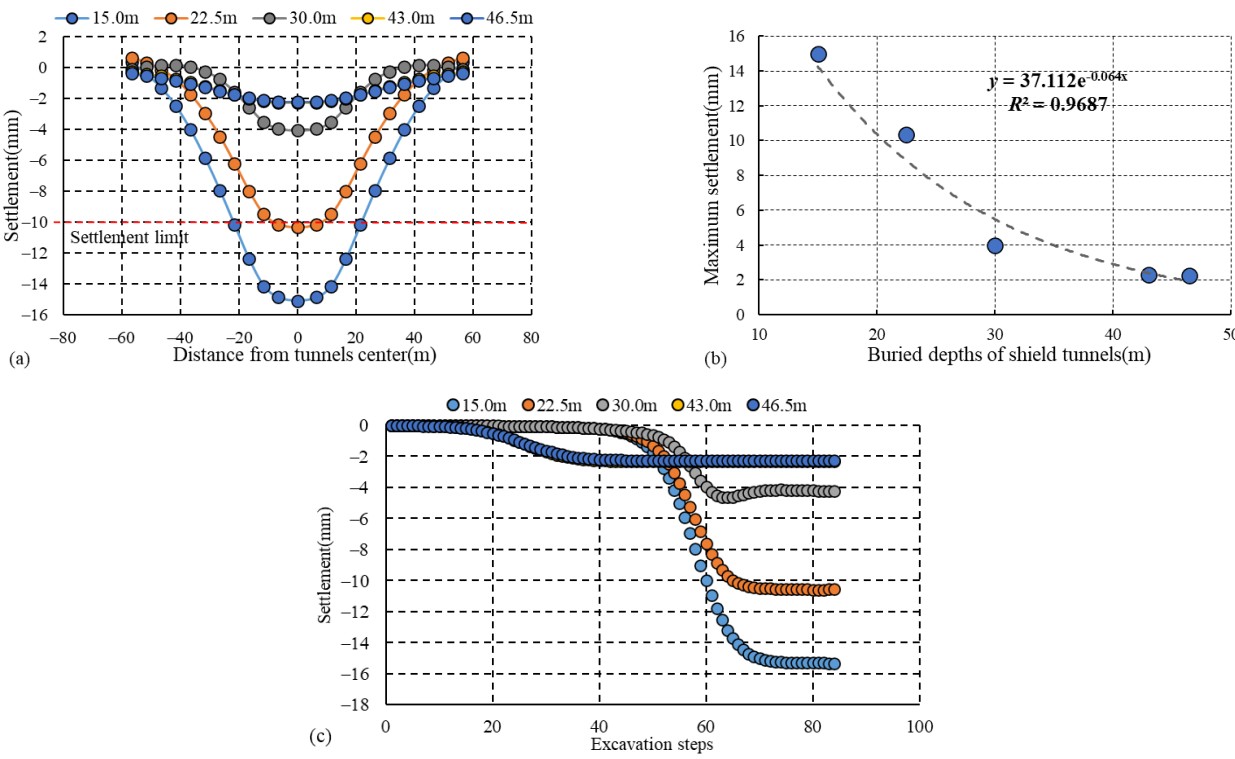

**Figure 14.** Influence of buried depth on ground surface settlement. (**a**) settlement trough. (**b**) the relationship between maximum settlement and tunnel burial depth. (**c**) settlement development.

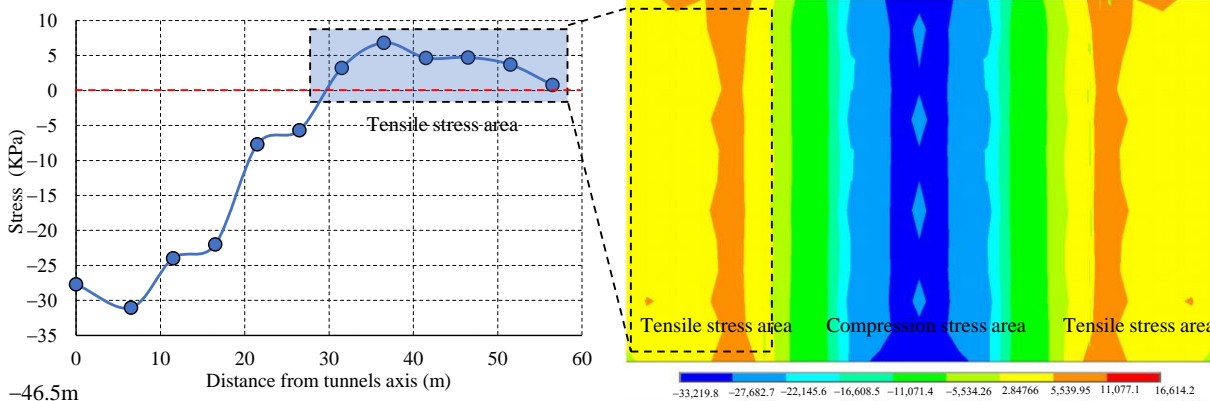

**Figure 15.** Stress in *x* direction of runway in transverse section (the buried depth = 46.5 m).

Case 2: the tunnel buried depth = 43.0 m.

The stress in the *x* direction of the runway in the transverse section when the buried depth = 43.0 m is shown in Figure 16. The pavement within 28.1 m of the shield tunnel axis was subjected to compressive stress. When the pavement was 56.5 m away from the tunnel axis, the stress basically dissipated to 0 kPa and the pavement 28.1–56.5 m away from the shield tunnel axis was subjected to tensile stress; the maximum tensile stress was 8.56 kPa.

Case 3: the tunnel buried depth = 30.0 m.

The stress in the *x* direction of the runway in the transverse section when the buried depth = 30.0 m is shown in Figure 17. The pavement within 23.8 m of the shield tunnel axis was subjected to compressive stress. When the pavement was 56.5 m away from the tunnel

axis, the stress basically dissipated to 0 kPa and the pavement 23.8–56.5 m away from the shield tunnel axis was subjected to tensile stress; the maximum tensile stress was 76.10 kPa.

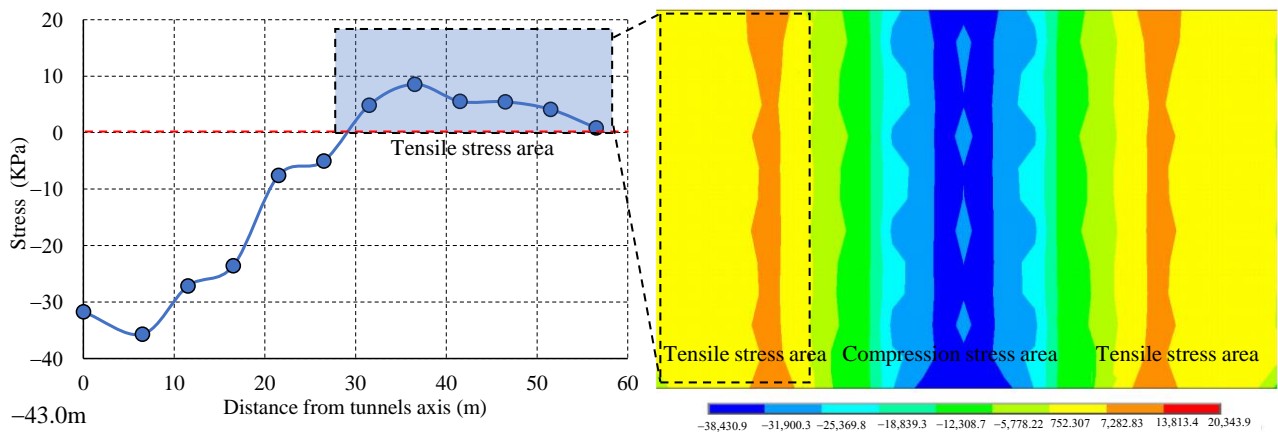

**Figure 16.** Stress in *x* direction of runway in transverse section (the buried depth = 43.0 m).

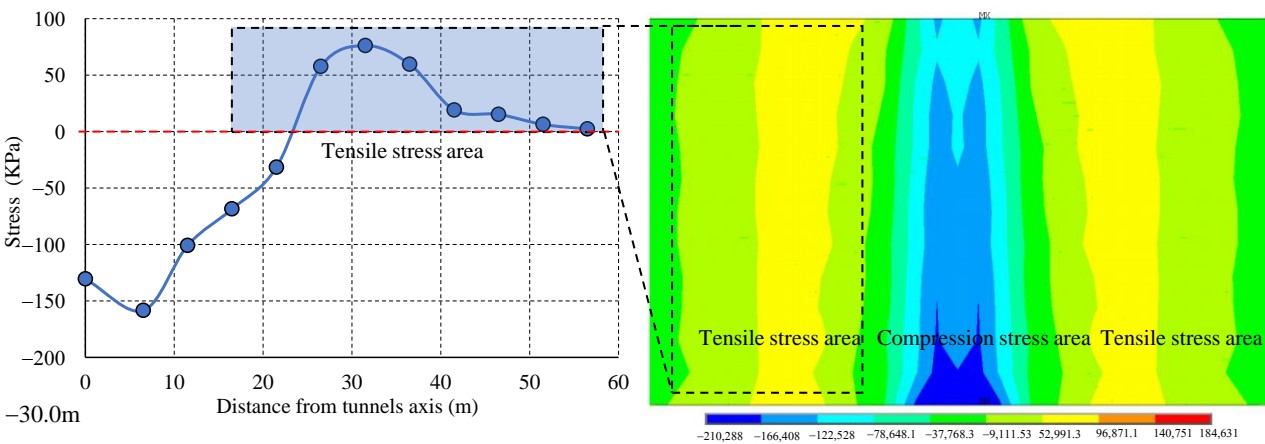

**Figure 17.** Stress in *x* direction of runway in transverse section (the buried depth = 30.0 m).

Case 4: the tunnel buried depth = 22.5 m.

The stress in the *x* direction of the runway in the transverse section when the buried depth = 22.5 m is shown in Figure 18. The pavement within 21.6 m of the shield tunnel axis was subjected to compressive stress. When the pavement was 56.5 m away from the tunnel axis, the stress basically dissipated to 0 kPa and the pavement 21.6–56.5 m away from the shield tunnel axis was subjected to tensile stress; the maximum tensile stress was 89.77 kPa.

Case 5: the tunnel buried depth = 15.0 m.

The stress in the *x* direction of the runway in the transverse section when the buried depth = 15.0 m is shown in Figure 19. The pavement within 20 m of the shield tunnel axis was subjected to compressive stress. When the pavement was 56.5 m away from the tunnel axis, the stress basically dissipated to 0 kPa and the pavement 20~56.5 m away from the shield tunnel axis was subjected to tensile stress; the maximum tensile stress was 95.01 kPa. Therefore, in the construction stage, the safety detection of this region should be considered to prevent the occurrence of runway cracks and other hidden dangers.

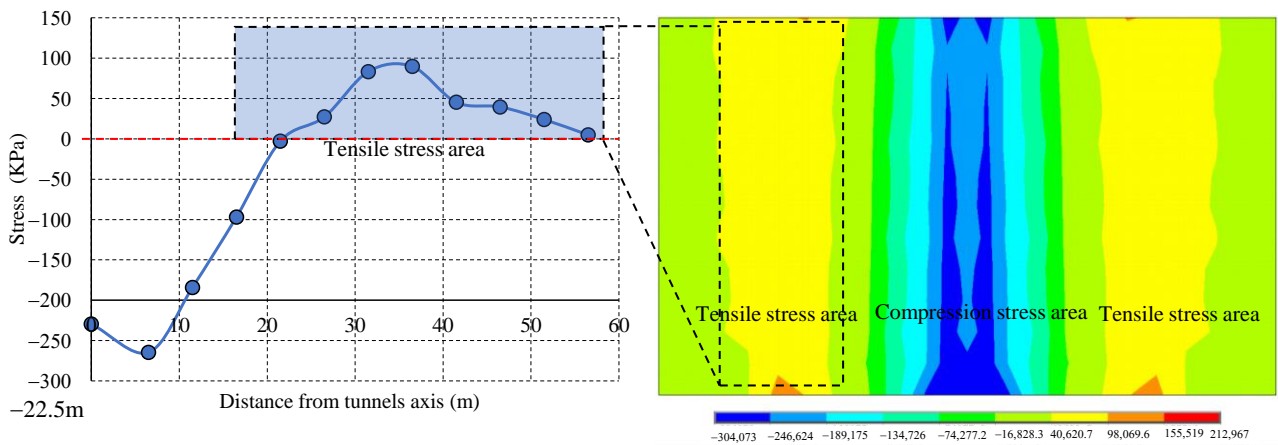

**Figure 18.** Stress in *x* direction of runway in transverse section (the buried depth = 22.5 m).

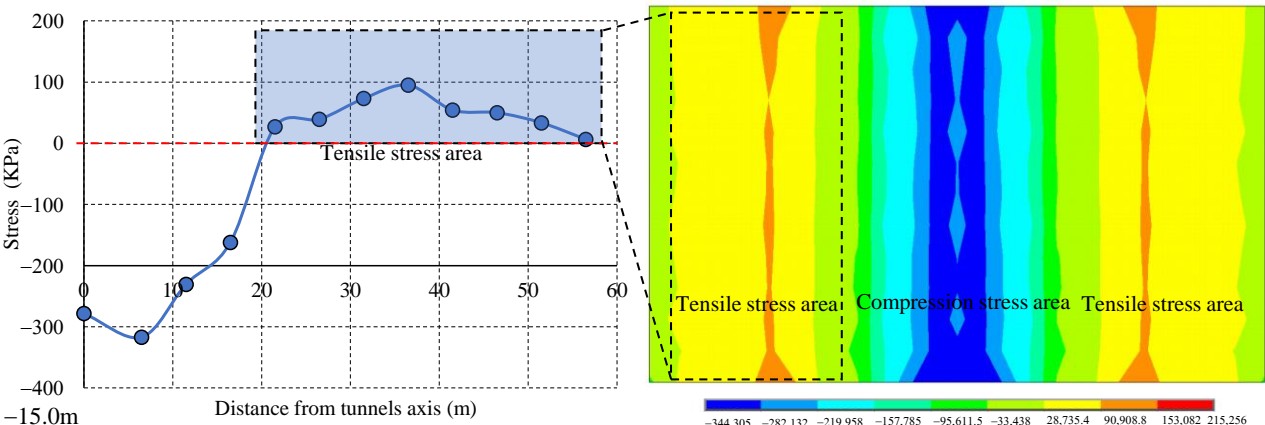

**Figure 19.** Stress in *x* direction of runway in transverse section (the buried depth = 15.0 m).

After the completion of the shield tunnel excavation, the uneven settlement of the overlying pavement structure will occur, which will cause sudden changes in the bending tensile stress of the pavement structure, leading to structural damage. The allowable tensile stress of the structural layer is the maximum fatigue stress when the pavement is subjected to repeated vehicle loads and reaches the critical failure state; this parameter can be used to characterize the flexural-tensile resistance of the structural layer. For the calculation of the allowable tensile stress of the pavement we referred to the provisions of specifications for "Specification for Design of Highway Asphalt Pavement (JTGD50-2006)" [37]. The allowable tensile stress $\sigma_R$ of the asphalt concrete pavement was defined as:

$$\sigma_R = \sigma_{SP}/K_s \tag{1}$$

where $\sigma_{SP}$ is the splitting strength (MPa) of the asphalt concrete and $K_S$ is the tensile strength coefficient, $Ks = 0.09 N_e^{0.22}/A_c$. Where $A_c$ is the road grade coefficient, 1.0 for expressway and Class I highway, and $N_e$ is the cumulative equivalent axle time in a lane within the design life; $1.8 \times 10^7$ was taken in this study. The allowable tensile stress of the pavement structure layer was calculated is 0.225 MPa. Considering the requirements for allowable tensile stress of the surface layer, the relationship between the tunnel buried depth and the pavement maximum tensile stress, the tensile stress area width was analysed, as shown in Figure 20. It can be concluded that there is a good linear relationship between the maximum tensile stress, the tensile stress area width and the tunnel buried depth; the judgment coefficients, $R^2$, were 0.9279 and 0.9852, respectively. With decreasing tunnel burial depth, the tensile stress area width and the maximum tensile stress value of pavement structure increased. Even if the tunnel buried depth was very shallow (15.0 m), the maximum

tensile stress of the asphalt pavement did not exceed the allowable tensile stress value; however, considering the influence of aircraft fatigue load after runway operation, the safety detection of the tensile area should be considered to prevent the occurrence of runway cracks and other hidden dangers.

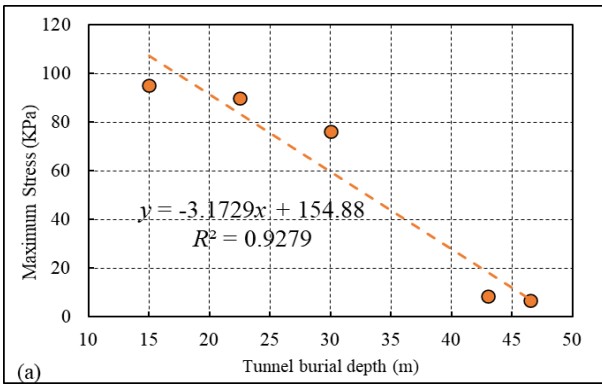

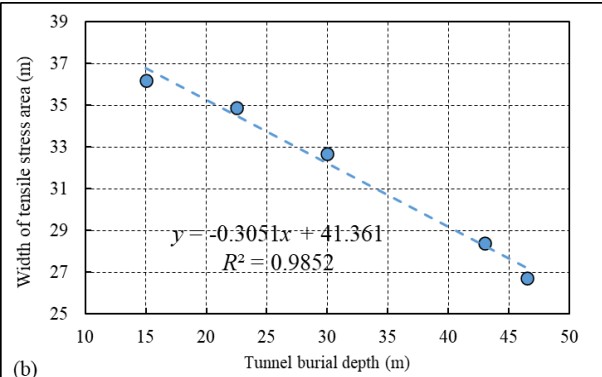

**Figure 20.** Influence of buried depth on ground surface settlement. (**a**) maximum stress vs. tunnel burial depth; (**b**) tensile stress area width vs. tunnel burial depth.

## 5. Conclusions

In this study, based on the case of the Chengdu Metro Line 10 undercrossing Shuangliu Airport in a sandy cobble region, a three-dimensional numerical model was established to investigate the ground settlement during the shield tunnel construction procedure. Then, the development law of the surface settlement during the double-line shield tunnel construction period was investigated and the influence of runway pavement on ground settlement at different depths was analysed. Subsequently, according to the stress analysis of the pavement structure, the most unfavourable regions of the runway pavement structure under the influence of tunnel excavation was determined. Finally, the influence of the tunnel burial depth on the ground surface settlement was studied. The main conclusions are as follows:

(1) When the double-line tunnel excavation was finished, the ground surface settlement curves at different sections induced by the double-line tunnels undercrossing were V-type settlement troughs instead of W-type settlement troughs. Among them, the maximum surface settlement at the pavement section was 1.96 mm and the maximum surface settlement at the sward section was 2.14 mm.

(2) When the shield machine advanced to the 40th loop, the settlement of the characteristic monitoring point at the sward surface reached the maximum value. When the shield machine advanced to the 60th loop, the settlement of C1 reached the maximum value.

(3) The existence of the runway pavement greatly limited the deformation of the surrounding soil; with increasing stratum depth, the effect degree of runway pavement on ground settlement decreased.

(4) With increasing tunnel buried depth, the maximum settlement of the surface centre point decreased continuously and the development time of the ground surface settlement was extended. It is suggested that the tunnel buried depth of the shield tunnel in this project should be no greater than 22.5 m.

(5) There were good linear relationships between the maximum tensile stress, the tensile stress area width and the tunnel buried depth. With decreasing tunnel buried depth, the tensile stress area width and the maximum tensile stress value of the pavement structure increased. Even if the maximum tensile stress of the asphalt pavement did not exceed the allowable tensile stress value, considering the influence of aircraft fatigue load after runway operation the safety detection of the tensile area should be considered.

The effect of shield construction on surface buildings has always been a complex problem in the engineering. This study only discussed the influence of tunnel buried depth and surface roads on the ground settlement of the operating airport in a sandy pebble region.

However, there are still some limitations, the influence of shield construction parameters, anisotropy of soil and the seepage of water needed be investigated in further research.

**Author Contributions:** Conceptualization, X.Z. and W.Q.; methodology, J.L.; software, W.L.; validation, X.Z.; formal analysis, J.L. and W.L.; investigation, J.L. and W.L.; resources, X.Z. and W.Q.; data curation, X.Z. and W.Q.; writing—original draft preparation, X.Z. and J.L.; writing—review and editing, X.Z. and J.L.; visualization, J.L. and W.L.; supervision, W.Q.; project administration, X.Z. and W.Q.; funding acquisition, X.Z. and W.Q. All authors have read and agreed to the published version of the manuscript.

**Funding:** This research received no external funding.

**Institutional Review Board Statement:** Not applicable.

**Informed Consent Statement:** Not applicable.

**Data Availability Statement:** The data that support the findings of this study are available from the first author upon reasonable request.

**Conflicts of Interest:** The authors declare that they have no known competing financial interests or personal relationships that could have appeared to influence the work reported in this paper.

## Abbreviations

| | |
|---|---|
| FP | First pavement |
| SP | Second pavement |
| LT | Left tunnel excavation |
| RT | Right tunnel excavation |
| $\gamma$ | Unit weight |
| c | Cohesion |
| $\phi$ | Internal friction angle |
| $v$ | Poisson's ratio |
| RQD | Rock quality designation |
| $\Phi$ | Diameter |
| L1 | Sections of runway midline |
| L2 | Sections of sward midline |
| RL1′ | Sections of runway midline when right-line tunnel excavation was finished |
| RL2′ | Sections of sward midline when right-line tunnel excavation was finished |
| LL1′ | Sections of runway midline when left-line tunnel excavation was finished |
| LL2′ | Sections of sward midline when left-line tunnel excavation was finished |
| L | Tunnel spacing |
| H | Tunnel burial depth |
| R | Tunnel radius |
| C1 | Sward characteristic monitoring point |
| D1 | Runway pavement characteristic monitoring point |
| $\sigma_R$ | Allowable tensile stress of asphalt concrete |
| $\sigma_{SP}$ | Splitting strength |
| $K_S$ | Tensile strength coefficient |
| $N_e$ | Cumulative equivalent axle time |
| $A_c$ | Road grade coefficient |

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
