# Peer review of "Settlement and Stress Characteristics of the Ground in the Project of a Double-Line Tunnel Undercrossing an Airport Runway in a Sandy Cobble Region"

_applsci, doi:10.3390/app122312498_

Round 1

Reviewer 1 Report

1.      The abstract is too short to explain the author’s effort. So the reviewer recommends that follow that pattern [background] -> [objective] -> [methodology] -> [results] -> [contribution].

2.      The overall introduction for this paper seems to be inappropriate. It was hard to understand about explaining the background, concept, and literature review. Especially, the literature review was a lack of contents. I recommend that the author need to add the literature review. The literature review section needs to be strengthened to provide a solid basis for the study.

3.      In Introduction, the literature review was a lack of contents. The reviewer recommends that the authors need to add the literature review to prove the research excellence rather than previous researches.

4.      In Method, I recommend, the author should insert the figure about the research process to enhance the understanding to the readers.

5.      The research gap, aim and contributions is not clear. The research aim should be clearly stated and more elaboration on the practical and theoretical implications of the findings from this study should be provided.

6.      In Conclusion, the author presents the research contribution, limitation and future work. However, It is not clear. The author should be explained in detailed and clear.

Author Response

Dear Editors and Reviewers:

Thank you for your letter and for the reviewers' comments concerning our manuscript entitled “Settlement and stress characteristic of ground in the project of double line tunnel undercrossing airport runway in sandy cobble region” (ID: applsci-1997749). In terms of the comments from the reviewers, the authors sincerely appreciate valuable comments received and all the changes have been marked as red in the revised manuscript.

The specific responses to the reviewer and revised manuscript are shown as the attachment.

Reviewer 2 Report

My comments are below after reading and evaluating the manuscript.

1.      Following the journals instruction for authors, the abstract must be more concise.

2.      The study must consider one aspect/objective and avoid multidimensional task.

3.      The model needs validation for the results authentication.

4.      A few study in my knowledge used CENTRIFUGE MODEL TESTS AND NUMERICAL ANALYSIS for the trough prediction.

Kim, Jonguk, et al. "Prediction of transverse settlement trough considering the combined effects of excavation and groundwater depression." Geomechanics and Engineering 15.3 (2018): 851-859.

An, J. B., Kang, S. J., Kim, J. J., Kim, K. Y., & Cho, G. C. (2021). A preliminary study for numerical and analytical evaluation of surface settlement due to EPB shield TBM excavation. Journal of Korean Tunnelling and Underground Space Association23(3), 183-198.

5.      Provide the project location in Map and also show the geological conditions/geological map of the area.

6.      Literature must be specific to this study.

7.      In table 1, use the brackets () only for the units and avoid “/”.

8.      Define all the variables and symbols used in the manuscript.

9.      Why 3D modelling? If 3D modelling, the the tunnel longitudinal profile and their discussion must be a part of this study. The results must be in line with the title.

10.   How the author’s selected the material properties? Any reference?

11.   How can the machine parameter (TBM chamber pressure, excavation rate, etc) affect the settlement?

12.   What will be the impact of groundwater depression on the trough.

Author Response

(The authors gave the same response as above.)

Reviewer 3 Report

1.  Change the abstract and write it down in constructed ways. 

2. Introduction should be changed and more information will be added. 

3. Grammer should be checked very carefully. 

4. Overall this paper will not accept. 

Author Response

(The authors gave the same response as above.)

Reviewer 4 Report

Manuscript ID: applsci-1997749

Title: Settlement and stress characteristic of ground in the project of

double line tunnel undercrossing airport runway in sandy cobble region 

Applied Sciences

The paper presents an interesting subject related to tunnels and ground settlement. The following notes were outlined:

1.     Page 1: The words “in sandy cobble region” are not necessary in the title.

2.     Page 2 - line 49 : What do you mean by “The influencing factors of the stratum deformation induced by shield tunneling are very complex and diverse” ?

3.     Page 3 – line 99: The following studies may be beneficial. You can also refer to them:

·        Fattah, M. Y., Hamood, M. J., Dawood, S. H., (2015), "Dynamic Response of a Lined Tunnel with Transmitting Boundaries", Earthquakes and Structures, Vol. 8, No. 1, pp. 275-304, DOI: http://dx.doi.org/10.12989/eas.2015.8.1.275, Techno-Press Journals.

·        Fattah, M. Y., Shlash, K. T., Salim, N. M., (2013), “Prediction of Settlement Trough Induced by Tunneling in Cohesive Ground”, Acta Geotechnica, DOI 10.1007/s11440-012-0169-4, Springer, Vol. 8, pp. 167–179.

·        Fattah, M. Y., Shlash, K. T., Salim, N. M., (2011), “Effect of Reduced Ko Zone on Time Dependent Analysis of Tunnels”, Advances in Civil Engineering, Vol. 2011, Article ID 963502, 12 pages, 2011. doi:10.1155/2011/963502, Hindawi Publishing Corporation. https://doi.org/10.1155/2011/963502.

·        Fattah, M. Y., Shlash, K. T., al-Soud, M. S., (2012), “Boundary Element Analysis of a Lined Tunnel Problem”, International Journal of Engineering, IJE TRANSACTIONS B: Applications Vol. 25, No. 2, (May 2012), pp.87-94.  doi: 10.5829/idosi.ije.2012.25.02b.02

·        Fattah, M. Y., Shlash, K. T., Salim, N. M., (2011), “Settlement Trough Due to Tunneling in Cohesive Ground”, Indian Geotechnical Journal, 41(2), 2011, 64-75. 

4.     In Table 1, please correct “density” to “unit weight”.

Mention the tests from which you obtained the soil parameters.

5.     What is the soil classification ?

6.     In Figure 3, describe the types of elements used.

7.     In Table 2, what is the material of lining ? the elastic modulus is very high.

8.     Page 7 - line 213: Give some details about Peck settlement trough.

9.     Aare the results of Figure 10 from finite element analysis ?

10.  Page 10 – line 284: How did you decide this depth ?

11.  Page 11 – line 307: What do you mean by “The first principal stress”? is it the major principal stress ?

12.  Page 13 – line 363: Define Ne and Ac.

Author Response

(The authors gave the same response as above.)

Reviewer 5 Report

This paper presents the case study of the Chengdu Metro Line crossing airport runways pavements, discussing in particular the influence of the depth of two tunnels on the settlements and on the stresses in the pavements. The case study is well presented and it has some interest; I have some concerns summarized below.

At line 128-129 you sat that “the pebble stratum is a typical mechanically unstable formation with a large gap between particles and no cohesion”, but in table 1 I see a cohesion value for the pebble soil.

What about the dilatancy angle?  I assume it is small for pebble soil, hence is it zero? What about the other layers?

About the finite elements: which is the order of the shape functions? In other words, are your elements linear or quadratic or else?

It is not included in the model, but it would be interesting to comment what the long-term behaviour of the settlement would be.

In Figure 9a there is a very small decrease of the settlement after excavation step 40. Can you explain why?

Author Response

(The authors gave the same response as above.)

Reviewer 6 Report

The reviewed paper is dedicated to the double line tunnel undercrossing Shuangliu airport runway in sandy cobble region of Chengdu Metro Line 10. The Authors prepared a three-dimensional numerical model to investigate the construction procedure of the shield tunnel. Secondly, the development law of the surface settlement during the double-line shield tunnel construction period was investigated, and the influence of the runway pavement and tunnel burial depth on the ground settlement was analysed. Finally, the vulnerable area of pavement structure was proposed. The Authors found the most unfavourable region of the runway pavement structure under the influence of tunnel excavation.

The reviewed paper is well organised, however there are two suggestions that could be taken into account. They are following:

- page 2, line 59: There is mentioned Michael et al. [27], while in the bibliography Kavvadas et al. is listed,

- Figure 9: quality of the figure should be improved.

In the reviewer's opinion, the paper is worth to publish.

Author Response

(The authors gave the same response as above.)

Round 2

Reviewer 1 Report

The authors addressed my comments, well.

Reviewer 2 Report

Remove the words "background", methodology, etc. from the abstract.